

**Characterization of the light absorbing properties, chromophores composition**
**and sources of brown carbon aerosol in Xi'an, Northwest China**
Wei Yuan[1,2], Ru-Jin Huang[1,3], Lu Yang[1], Jie Guo[1], Ziyi Chen[4], Jing Duan[1,2], Meng Wang[1,2], Ting
Wang[1,2], Haiyan Ni[1], Yongming Han[1], Yongjie Li[5], Qi Chen[6], Yang Chen[7], Thorsten Hoffmann[8],
Colin O'Dowd[9]
[1]State Key Laboratory of Loess and Quaternary Geology, Center for Excellence in Quaternary
Science and Global Change, Chinese Academy of Sciences, and Key Laboratory of Aerosol
Chemistry & Physics, Institute of Earth Environment, Chinese Academy of Sciences, Xi'an
710061, China
[2]University of Chinese Academy of Sciences, Beijing 100049, China
[3]Institute of Global Environmental Change, Xi'an Jiaotong University, Xi'an 710049, China
[4]Royal School of Mines, South Kensington Campus, Imperial College London, Exhibition
Road, London SW7 3RW, United Kingdom
[5]Department of Civil and Environmental Engineering, Faculty of Science and Technology,
University of Macau, Taipa, Macau 999078, China
[6]State Key Joint Laboratory of Environmental Simulation and Pollution Control, College of
Environmental Sciences and Engineering, Peking University, Beijing 100871, China
[7]Chongqing Institute of Green and Intelligent Technology, Chinese Academy of Sciences,
Chongqing 400714, China
[8]Institute of Inorganic and Analytical Chemistry, Johannes Gutenberg University Mainz,
Duesbergweg 10−14, Mainz 55128, Germany
[9]School of Physics and Centre for Climate and Air Pollution Studies, Ryan Institute, National
University of Ireland Galway, University Road, Galway H91CF50, Ireland
*Correspondence to*: Ru-Jin Huang (rujin.huang@ieecas.cn)
**Abstract**
The impact of brown carbon aerosol (BrC) on the Earth's radiative forcing balance has


been widely recognized but remains uncertain, mainly because the relationships among BrC
sources, chromophores, and optical properties of aerosol are poorly understood. In this work,
the light absorption properties and chromophore composition of BrC were investigated for
samples collected in Xi'an, Northwest China from 2015 to 2016. Both absorption Ångström
exponent and mass absorption efficiency show distinct seasonal differences, which could be
attributed to the differences in sources and chromophore composition of BrC. Three groups of
light-absorbing organics were found to be important BrC chromophores, including those show
multiple absorption peaks at wavelength > 350 nm (12 polycyclic aromatic hydrocarbons and
their derivatives) and those show single absorption peak at wavelength < 350 nm (10
nitrophenols and nitrosalicylic acids and 3 methoxyphenols). These measured BrC
chromophores show distinct seasonal differences and contribute on average about 1.1% and 3.3%
of light absorption of methanol-soluble BrC at 365 nm in summer and winter, respectively,
about 7 and 5 times higher than the corresponding carbon mass fractions in total organic carbon.
The sources of BrC were resolved by positive matrix factorization (PMF) using these
chromophores instead of commonly used non-light absorbing organic markers as model inputs.
Our results show that in spring vehicular emissions and secondary formation are major sources
of BrC (~70%), in fall coal combustion and vehicular emissions are major sources (~70%), in
winter biomass burning and coal combustion become major sources (~80%), while in summer
secondary BrC dominates (~60%).
**1 Introduction**
Brown carbon (BrC) is an important component of atmospheric aerosol particles and has
significant effects on radiative forcing and climate (Feng et al., 2013; Laskin et al., 2015; Zhang
et al., 2017a). BrC can efficiently absorb solar radiation and reduce the photolysis rates of
atmospheric radicals (Jacobsan, 1999; Li et al., 2011; Mok et al., 2016), which ultimately
influences the atmospheric photochemistry process, the formation of secondary organic aerosol
(SOA), and therefore the regional air quality (Mohr et al., 2013; Laskin et al., 2015; Moise et
al., 2015). In addition, some components in BrC, such as nitrated aromatic compounds (NACs)
(Teich et al., 2017; Wang et al., 2018) and polycyclic aromatic hydrocarbons (PAHs)
(Samburova et al., 2016; Huang et al., 2018), have adverse effects on human health. The



significant effects of BrC on environment, climate, air quality and living things call for more
studies to understand its chemical characteristics, sources and the links with optical properties.

Investigating the chemical composition of BrC at molecular level is necessary, because

even small amounts of compounds can have a significant effect on the light absorption
properties of BrC and profound atmospheric implication (Mohr et al., 2013; Zhang et al., 2013;
Teich et al., 2017; Huang et al., 2018). A number of studies have investigated the BrC
composition at molecular level (Mohr et al., 2013; Zhang et al., 2013; Chow et al., 2015;
Samburova et al., 2016; Lin et al., 2016, 2017, 2018; Teich et al., 2017; Huang et al., 2018; Lu
et al., 2019). For example, Zhang et al. (2013) measured 8 NACs in Los Angeles and found that
they contributed about 4% of water-soluble BrC absorption at 365 nm. Huang et al. (2018)
measured 18 PAHs and their derivatives in Xi'an and found that they accounted for on average
~1.7% of the overall absorption of methanol-soluble BrC. A state-of-the-art high performance
liquid chromatography-photodiode array-high resolution mass spectrometry (HPLC-PDA-
HRMS) was applied to investigate the elemental composition of BrC chromophores in biomass
burning aerosol (Lin et al., 2016, 2017, 2018). Despite these efforts, the molecular composition
of atmospheric BrC still remains largely unknown due to its complexity in emission sources
and formation processes.

Field observations and laboratory studies show that BrC has various sources, including

primary emissions such as combustion and secondary formation from various atmospheric
processes (Laskin et al., 2015). Biomass burning, including forest fires and burning of crop
residues, is considered as the main source of BrC (Teich et al., 2017; Lin et al., 2017). Coal
burning and vehicle emissions are also important primary sources of BrC (Yan et al., 2017; Xie
et al., 2017). Secondary BrC is produced through multiple-phase reactions occurring in or
between gas phase, particle phase, and cloud droplets. For example, nitrification of aromatic
compounds (Harrison et al., 2005; Lu et al., 2011), oligomers of acid-catalyzed condensation
of hydroxyl aldehyde (De Haan et al., 2009; Shapiro et al., 2009), and reaction of ammonia
($NH_3$) or amino acids with carbonyls (De Haan et al., 2011; Nguyen et al., 2013; Flores et al.,
2014) can all produce BrC. Condensed phase reactions and aqueous-phase reactions have also
been found to be important formation pathways for secondary BrC in ambient air (Gilardoni et



al., 2016). In addition, atmospheric aging processes can lead to either enhancement or bleaching
of the BrC absorption (Lambe et al., 2013; Lee et al., 2014; Zhong and Jang, 2014), further
challenging the characterization of BrC.
As the starting point of the Silk Road, Xi'an is an important inland city in northwestern
China experiencing severe particulate air pollution, especially during heating period with
enhanced coal combustion and biomass burning activities (Wang et al., 2016; Ni et al., 2018).
In this study, we performed spectroscopic measurement and chemical analysis of PM$_{2.5}$ filter
samples in Xi'an to investigate: 1) seasonal variations in the light absorption properties and
chromophore composition of BrC, and their relationships; 2) sources of BrC in different seasons
based on positive matrix factorization (PMF) model with light-absorbing organic markers as
input species.
**2 Experimental**
**2.1 Aerosol sampling**
A total of 112 daily ambient PM$_{2.5}$ filter samples were collected on pre-baked (780 °C, 3
h) quartz-fiber filters (20.3 × 25.4 cm, Whatman, QM-A) in November-December 2015, April-
May, July, October-November 2016, representing winter, spring, summer and fall, respectively.
Filter samples were collected using a Hi-Vol PM$_{2.5}$ air sampler (Tisch, Cleveland, OH) at a flow
rate of 1.05 m$^3$ min$^{-1}$ on the roof (~10 m above ground level, 34.22°N, 109.01°E) of the Institute
of Earth Environment, Chinese Academy of Sciences, which was surrounded by residential
areas without large industrial activities. After collection, the filter samples were wrapped in
baked aluminum foils and stored in a freezer (-20 °C) until further analysis.
**2.2 Light absorption measurement**
One punch of loaded filter (0.526 cm$^2$) was taken from each sample and sonicated for 30
minutes in 10 mL of ultrapure water (> 18.2 MΩ · cm) or methanol (J. T. Baker, HPLC grade).
The extracts were then filtered with a 0.45 μm PTFE pore syringe filter to remove insoluble
materials. The light absorption spectra of water-soluble and methanol-soluble BrC were
measured with an UV-Vis spectrophotometer (300-700 nm) equipped with a liquid waveguide
capillary cell (LWCC-3100, World Precision Instrument) following the method by Hecobian et



al. (2010). The measured absorption data can be converted to the absorption coefficient by
equation (1):

$$Abs_\lambda = (A_\lambda - A_{700})\frac{V_l}{V_a \times L} \times \ln(10) \qquad (1)$$

where $A_{700}$ is the absorption at 700 nm, serving as a reference to account for baseline drift, $V_l$
is the volume of water or methanol that the filter was extracted into, $V_a$ is the volume of sampled
air, L is the optical path length (0.94 m). A factor of $\ln(10)$ is used to convert the log base-10
(recorded by UV-Vis spectrophotometer) to natural logarithm to provide base-e absorption
coefficient. The absorption coefficient of water-soluble or methanol-soluble organics at 365 nm
($Abs_{365}$) is used to represent water-soluble or methanol-soluble BrC absorption, respectively.

The mass absorption efficiency (MAE) of BrC in the extracts can be calculated as:

$$MAE_\lambda = \frac{Abs_\lambda}{M} \qquad (2)$$

where M ($\mu gC\ m^{-3}$) is the concentration of water-soluble organic carbon (WSOC) for water
extracts or methanol-soluble organic carbon (MSOC) for methanol extracts. Note that organic
carbon (OC) is often used to replace MSOC because direct measurement of MOSC is
technically difficult and many studies have shown that most of OC ($\sim 90\%$) can be extracted
by methanol (Chen and Bond, 2010; Cheng et al., 2016; Xie et al., 2019).

The wavelength-dependent light absorption of chromophores in solution, termed as

absorption Ångström exponent (AAE), can be described as:

$$Abs_\lambda = K \cdot \lambda^{-AAE} \qquad (3)$$

where K is a constant related to the concentration of chromophores and AAE is calculated by
linear regression of $\log Abs_\lambda$ versus $\log \lambda$ in the wavelength range of 300-410 nm.

**2.3 Chemical analysis**

OC was measured with a thermal/optical carbon analyzer (DRI, model 2001) following

the IMPROVE-A protocol (Chow et al., 2011). WSOC was measured with a TOC/TN analyzer
(TOC−L, Shimadzu, Japan) (Ho et al., 2015).

Organic compounds listed in Table S1 were analyzed with a gas chromatograph-mass

spectrometer (GC-MS). The concentrations of NACs were analyzed following the method by
Al-Naiema and Stone (2017). Briefly, a quarter of 47 mm filter sample was ultrasonically





extracted with 2 mL of methanol for 15 minutes and repeated three times. 4-Nitrophenol-
2,3,5,6-d$_4$ was added as an internal standard before extraction to correct for potential loss of
analytes during the extraction process. The extracts were filtered with a 0.45 μm PTFE syringe
filter and then evaporated with a rotary evaporator to ~1 mL and dried with a gentle stream of
nitrogen. Then, 50 μL of N,O-bis(trimethylsilyl)trifluoroacetamide (BSTFA-TMCS; Fluka
Analytical 99%) and 10 μL of pyridine were added. The mixture was heated for 3 h at 70 ºC for
silylation. After reaction, 140 μL of n-hexane were added to dilute the derivatives. Finally, 2 μL
aliquot of the derivatized extracts were introduced into the GC-MS, which was equipped with
a DB-5MS column, electron impact (EI) ionization source (70 eV), and a GC inlet of 280 ºC.
The GC oven temperature was held at 50 ºC for 2 min, increased from 50 ºC to 120 ºC at a rate
of 15 ºC min$^{-1}$, then further increased from 120 ºC to 300 ºC at a rate of 10 ºC min$^{-1}$ for a total
running time of 25 min. The concentrations of PAHs and its oxygenated derivatives,
methoxyphenols (MOPs), levoglucosan, hopanes and phthalic acid were analyzed following
methods described by Wang et al. (2006).

**2.4 Source apportionment of BrC**

Source apportionment of methanol-soluble BrC was performed using positive matrix
factorization (PMF) as implemented by the multilinear engine (ME-2; Paatero, 1997) via the
Source Finder (SoFi) interface written in Igor Wavemetrics (Canonaco et al., 2013). Abs$_{365,MSOC}$
and those light-absorbing species including fluoranthene (FLU), pyrene (PYR), chrysene
(CHR), benzo(a)anthracene (BaA), benzo(a)pyrene (BaP), benzo(b)fluoranthene (BbF),
benzo(k)fluoranthene (BkF), indeno[1,2,3-cd]pyrene (IcdP), benzo(ghi)perylene (BghiP), 9,10-
anthracenequinone (9,10-AQ), benzanthrone (BEN), benzo[b]fluoren-11-one (BbF11O),
vanillic acid, vanillin and syringyl acetone were used as model inputs, together with some non-
light absorbing markers, i.e., phthalic acid, hopanes (17α(H),21β(H)-30-norhopane,
17α(H),21β(H)-hopane,        17α(H),21β(H)-(22S)-homohopane,        17α(H),21β(H)-(22R)-
homohopane, referred to as HP1-HP4, respectively), picene, and levoglucosan. The input data
include species concentrations and uncertainties. The method detection limits (MDLs),
calculated as three times of the standard deviation of the blank filters, were used to estimate
species-specific uncertainties, following Liu et al. (2017). Furthermore, for a clear separation





of sources profiles, the contribution of corresponding markers was set to 0 in the sources
unrelated to the markers (see Table S2).

## 3 Results and discussion

### 3.1 Light absorption properties of water- and methanol-soluble BrC

Fig. 1 shows the temporal profiles of $Abs_{365}$ of water- and methanol-soluble BrC, together
with the concentrations of WSOC and OC (representing MSOC). They all show similar
seasonal variations with the highest average in winter, followed by fall, spring and summer (see
Table S3). WSOC contributed annually $54.4 \pm 16.2\%$ of the OC mass, with the highest
contribution in summer ($66.1 \pm 15.5\%$) and the lowest contribution in winter ($45.1 \pm 10.2\%$).
The higher WSOC fraction in OC during summer may be related to biomass burning emissions
and SOA formation which produce more WSOC (Ram et al., 2012; Yan et al., 2015). The lower
WSOC fractions in OC during winter could be attributed to enhanced emissions from coal
combustion and motor vehicles which produce a large fraction of water-insoluble organics (Dai
et al., 2015; Daellenbach et al., 2016; Yan et al., 2017). $Abs_{365,MSOC}$ is approximately 2 times
(range 1.7-2.3) higher than $Abs_{365,WSOC}$, which is similar to the results measured in Beijing
(Cheng et al., 2016), southeastern Tibetan Plateau (Zhu et al., 2018), Gwangju, Korea (Park et
al., 2018) and the Research Triangle Park, USA (Xie et al., 2019), indicating that the optical
properties of BrC could be largely underestimated when using water as the extracting solvent.
In Fig. S1 we summarized those previously reported $Abs_{365,WSOC}$ (as $Abs_{365,MSOC}$ was not
commonly measured in many previous studies) values at different sites in Asian urban and
remote areas and the US. $Abs_{365,WSOC}$ is significantly higher in most Asian urban regions than
in the Asian remote sites and the US, and show clear seasonal variations. The high light
absorption of BrC in Asian urban regions, especially during winter, may have important effects
on regional climate and radiation forcing (Park et al., 2010; Laskin et al., 2015). As discussed
in Feng et al. (2013), the average global climate forcing of BrC was estimated to be 0.04-0.11
$W\ m^{-2}$ and above $0.25\ W\ m^{-2}$ in urban sites of south and east Asia regions, which is about 25%
of the radiative forcing of black carbon (BC, $1.07\ W\ m^{-2}$). Thus, to further understand the
influence of BrC on regional radiation forcing, it is essential to identify and quantify the sources



of BrC in Asia.

The seasonal averages of AAE of water-soluble BrC were between 5.32 and 6.15 without

clear seasonal trend (see Table S3). The seasonal averages of AAE of methanol-soluble BrC
were relatively lower than those of water-soluble BrC, ranging from 4.45 to 5.18 which is
similar to the results in Los Angeles Basin (Zhang et al., 2013) and Gwangju, Korea (Park et
al., 2018). This is because methanol can extract more compounds with high conjugation degree
and strong light-absorbing capability (e.g., PAHs) at longer wavelength (> 350 nm). The AAE
values of water-soluble BrC (as AAE of methanol-soluble BrC was not commonly measured in
many previous studies) in urban, rural and remote regions show a large difference (see Fig. 2a),
typically with much lower AAE values in urban regions than those in rural and remote regions,
indicating the difference in sources and chemical composition of chromophores. The urban
regions are mainly affected by anthropogenic emissions. Therefore, urban BrC may contain a
large amount of aromatic chromophores with high conjugation degree, which absorb light at a
longer wavelength and have lower AAE values (Lambe et al., 2013; Wang et al., 2018).

The average $MAE_{365}$ values of water- and methanol-soluble BrC show large seasonal

variations, with higher values in winter (1.85 and 1.50 $m^2\,gC^{-1}$, respectively) and fall (1.18 and
1.52 $m^2\,gC^{-1}$), and lower values in spring (1.01 and 0.79 $m^2\,gC^{-1}$) and summer (0.91 and 1.21
$m^2\,gC^{-1}$). Such large seasonal differences indicate seasonal difference in BrC sources, as
discussed below. Compared to previous studies (Fig. 2b), the average values of $MAE_{365,WSOC}$
are obviously higher in urban sites than in rural and remote sites. The higher $MAE_{365,WSOC}$
values in urban regions is likely associated with enhanced anthropogenic emissions from e.g.,
coal combustion and biomass burning, and the lower $MAE_{365,WSOC}$ values in rural and remote
regions could be attributed to biogenic sources or aged secondary BrC (Lei et al., 2018; Xie et
al., 2019).
**3.2 Chemical characterization of the BrC chromophores**

Given the complexity in emission sources and formation processes, the molecular

composition of atmospheric BrC remains largely unknown. PAHs, NACs and MOPs have
recently been found as major chromophores in biomass burning-derived BrC (Lin et al., 2016,
2017, 2018). However, these compounds can also be directly emitted by coal combustion and



motor vehicle or formed by secondary reactions (Harrison et al., 2005; Iinuma et al., 2010; Liu
et al., 2017; Wang et al., 2018; Lu et al., 2019), making source attribution of atmospheric BrC
more challenging. To obtain the exact molecular composition of BrC chromophores and
understand the influence of a specific chromophore on BrC optical property, we measured the
light absorption characteristics of available chromophore standards including 12 PAHs, 10
NACs and 3 MOPs, and quantified their concentrations in $PM_{2.5}$ samples with GC-MS. The
light absorption contribution of individual chromophores to that of methanol-soluble BrC in the
wavelength range of 300-500 nm was estimated according to its concentration and mass
absorption efficiency (see Supplementary). Fig. 3 shows the contribution of carbon content in
identified BrC chromophores to the total OC mass. They all show obvious seasonal variations
with the highest values in winter and lowest in summer. The seasonal difference can be up to a
factor of 5-6. The contribution of PAHs ranged from 0.12% in summer to 0.47% in winter,
NACs from 0.02% in summer to 0.13% in winter, and MOPs from 0.01% in summer to 0.06%
in winter. It should be noted that NACs are dominated by 4-nitrophenol and 4-nitrocatechol in
spring, fall and winter, but by 4-nitrophenol and 5-nitrosalicylic acid in summer. The difference
is likely due to enhanced summertime formation of 5-nitrosalicylic acid, which is more oxidized
than other nitrated phenols measured in this study (Wang et al., 2018).

The seasonally averaged contributions of PAHs, NACs, MOPs and total measured

chromophores to light absorption of methanol-soluble BrC between 300 to 500 nm are shown
in Fig. 4. They show large seasonal variations and wavelength dependence. Specifically, PAHs
made the largest contribution to BrC light absorption in autumn, followed by winter, spring and
summer, and show two large absorption peaks at about 365 nm and 380 nm, which are mainly
associated with the absorption of BaP, BghiP, IcdP, FLU, BkF and BaA (see Fig. S2). Compared
to PAHs, NACs show the largest contribution in winter, followed by fall, spring and summer,
and exhibit only one absorption peak at about 320 nm in spring and summer and at about 330
nm in fall and winter. The red shift in the absorption peak could be attributed to the increase of
contributions from 4-nitrocatechol, 4-methyl-5nitrocatechol and 3-methyl-5-nitrocatechol
which have absorption peak at about 330-350 nm (see Fig. S2). Different from PAHs and NACs,
MOPs contribute the most in winter, followed by spring, fall and summer, and only show one



absorption peak at about 310 nm. The difference in light absorption contributions of different
chromophores in different seasons reflects the difference in sources, emission strength and
atmospheric formation processes.
The total contributions of PAHs, NACs and MOPs to the light absorption of methanol-
soluble BrC at 365 nm ranged from 1.05% (summer) to 3.26% (winter) (see Table 1). The
average contribution of PAHs to the BrC light absorption at 365 nm was 0.97% in summer (the
lowest) and 2.69% in fall (the highest), the contribution of NACs was 0.09% in summer and
0.82% in winter, and the contribution of MOPs was 0.006% in summer and 0.024% in winter.
The low contributions of these measured chromophores to the light absorption of methanol-
soluble BrC are consistent with previous studies. For example, Huang et al. (2018) measured
18 PAHs and their derivatives, which on average contributed ~1.7% of the overall absorption
of methanol-soluble BrC in Xi'an. Mohr et al. (2013) estimated the contribution of five NACs
to particulate BrC light absorption at 370 nm to be ~4% in Detling, UK. Zhang et al. (2013)
measured eight NACs, which accounted for ~4% of water-soluble BrC absorption at 365 nm in
Los Angeles. Teich et al. (2017) determined eight NACs during six campaigns at five locations
in summer and winter, and founded that the mean contribution of NACs to water-soluble BrC
absorption at 370 nm ranged from 0.10% to 1.25% under acidic conditions and from 0.13% to
3.71% under alkaline conditions. Slightly different from these previous studies, we investigated
the contributions of three groups of chromophores with different light-absorbing properties to
the light absorption of BrC, and provided further understanding in the relationships between
optical properties and chemical composition of BrC in the atmosphere. For example, vanillin,
which has negligible contribution to BrC light absorption at 365 nm, can produce secondary
BrC through oxidation and thus enhance the light absorption by a factor of 5-7 (Li et al., 2014;
Smith et al., 2016). The contribution of PAHs to the light absorption of methanol-soluble BrC
at 365 nm was 5-13 times that of their mass fraction of carbon in OC, 6-9 times for NACs, and
0.4-0.7 times for MOPs (4-8 times at 310 nm for MOPs). These results further demonstrate that
even a small amount of chromophores can have a disproportionately high impact on the light
absorption properties of BrC, and that the light absorption of BrC is likely determined by a
number of chromophores with strong light absorption ability (Kampf et al., 2012; Teich et al.,



2017).

**3.3 Sources of BrC**

Two approaches have been used to quantify the sources of BrC, including multiple linear

regression and receptor models such as PMF. For example, Washenfelder et al. (2015) utilized
multiple linear regression to determine the contribution of individual OA factors resolved by
PMF to OA light absorption in the southeastern America. Moschos et al. (2018) combined the
time series of PMF-resolved OA factors with the time series of light absorption of water-soluble
OA extract as model inputs to quantify the sources of BrC in Magadino and Zurich, Switzerland.
Xie et al. (2019) quantified the sources of BrC in southeastern America using $Abs_{365}$, elemental
carbon (EC), OC, WSOC, isoprene sulfate ester, monoterpene sulfate ester, levoglucosan and
isoprene SOA tracers as PMF model inputs. However, it should be noted that previous studies
mainly rely on the correlation between measured light absorption and organic tracers that do
not contain a BrC chromophore, and therefore may lead to bias in BrC source apportionment.
To better constrain the sources of BrC (i.e., contribution to $Abs_{365,MSOC}$), we used BrC
chromophores as PMF model inputs. The inputs include vanillic acid, vanillin, and syringyl
acetone for BrC from biomass burning, and FLU, PYR, CHR, BaA, BaP, BbF, BkF, IcdP, BghiP,
9,10AQ, BEN, and BbF11O for BrC from incomplete combustion. In addition, we included
non-light absorbing levoglucosan for biomass burning, phthalic acid for secondary BrC,
hopanes for vehicle emission and picene for coal burning in the model inputs.

Four factors were resolved, including vehicle emission, coal burning, biomass burning and

secondary formation. The profile of each factor is shown in Fig. S3. The first factor is
characterized by a high contribution of phthalic acid, a tracer of secondary formation of OA.
The second factor is dominated by hopanes, mainly from vehicular emissions. The third factor
is characterized by high contributions of PI, BaP, BbF, BkF, IcdP, BghiP, mainly from coal
combustion emissions, while the fourth factor has high contributions of levoglucosan, vanillic
acid, vanillin, syringyl acetone from biomass burning emissions. The seasonal difference in
relative contribution of each factor to BrC light absorption is shown in Fig. 5. In spring,
vehicular emissions (34%) and secondary formation (37%) were the main contributors to BrC
and coal combustion also had a relatively large contribution (29%). In summer, secondary



formation constituted the largest fraction (~60%), mainly due to enhanced photochemical
formation of secondary BrC. In fall, vehicular emissions (38%), coal combustion (29%) and
biomass burning (22%) all had significant contributions to BrC. In winter, coal combustion
(44%) and biomass burning (36%) were the main contributors, due to emissions from
residential biomass burning (wood and crop residues) and coal combustion for heating. Such
large seasonal difference in emission sources and atmospheric processes of BrC indicates that
more studies are required to better understand the relationship between chemical composition,
formation processes, and light absorption properties of BrC.

**4 Conclusion**

The light absorption properties of water- and methanol-soluble BrC in different seasons
were investigated in Xi'an. The light absorption coefficient of methanol-soluble BrC was
approximately 2 times higher than that of water-soluble BrC at 365 nm, and had an average
$MAE_{365}$ value of $1.27 \pm 0.46$ m$^2$ gC$^{-1}$. The average $MAE_{365}$ value of water-soluble BrC was 1.19
$\pm 0.51$ m$^2$ gC$^{-1}$, which is comparable to those in previous studies at urban sites but higher than
those in rural and remote areas. The seasonally averaged AAE values of water-soluble BrC
ranged from 5.32 to 6.15, which are higher than those of methanol-soluble BrC (between 4.45
and 5.18). In combination with previous studies, we found that AAE values of water-soluble
BrC were much lower in urban regions than those in rural and remote regions. The difference
of optical properties of BrC in different regions could be attributed to the difference in sources
and chemical composition of BrC chromophores. The contributions of 12 PAHs, 10 NACs and
3 MOPs to the light absorption of methanol-soluble BrC were determined and showed large
seasonal variations. Specifically, the total contribution to methanol-soluble BrC light absorption
at 365 nm ranged from 1.1% to 3.3%, which is 5-7 times higher than their carbon mass fractions
in total OC. This result indicates that the light absorption of BrC is likely determined by an
amount of chromophores with strong light absorption ability. Four major sources of methanol-
soluble BrC were identified, including secondary formation, vehicle emission, coal combustion
and biomass burning. On average, secondary formation and vehicular emission were the main
contributors of BrC in spring (~70%). Vehicular emission (38%), coal burning (29%) and
biomass burning (22%) all contributed significantly to BrC in fall. Coal combustion and



biomass burning were the major contributors in winter (~80%), and secondary formation was
the predominant source in summer (~60%). The large variations of BrC sources in different
seasons suggest that more studies are needed to understand the seasonal difference in chemical
composition, formation processes, and light absorption properties of BrC, as well as their
relationships.
**5 Abbreviations of organics**
**PAHs (Polycyclic Aromatic Hydrocarbons)**
BaA                    Benzo(a)anthracene
BaP                    Benzo(a)pyrene
BbF                    Benzo(b)fluoranthene
BbF11O                 Benzo[b]fluoren-11-One
BEN                    Benzanthrone
BghiP                  Benzo(ghi)perylene
BkF                    Benzo(k)fluoranthene
CHR                    Chrysene
FLU                    Fluoranthene
IcdP                   Indeno[1,2,3-cd]pyrene
PYR                    Pyrene
9,10AQ                 9,10-Anthracenequinone
**NACs (Nitrated Aromatic Compounds)**
2M4NP                  2-Methyl-4-Nitrophenol
2,6DM4NP               2,6-Dimethyl-4-Nitropheol
3M4NP                  3-Methyl-4-Nitrophenol
3M5NC                  3-Methyl-5-Nitrocatechol
3NSA                   3-Nitrosalicylic Acid
4M5NC                  4-Methyl-5-Nitrocatechol
4NC                    4-Nitrocatechol
4NP                    4-Nitrophenol



| | | |
|---|---|---|
| 371 | 4N1N | 4-Nitro-1-Naphthol |
| 372 | 5NSA | 5-Nitrosalicylic Acid |

**MOP (Methoxyphenols)**

| | | |
|---|---|---|
| 374 | SyA | Syringyl Acetone |
| 375 | VaA | Vanillic Acid |
| 376 | VAN | Vanillin |

**Hopanes**

| | | |
|---|---|---|
| 378 | HP1 | $17\alpha(H),21\beta(H)$-30-Norhopane |
| 379 | HP2 | $17\alpha(H),21\beta(H)$-Hopane |
| 380 | HP3 | $17\alpha(H),21\beta(H)$-(22S)-Homohopane |
| 381 | HP4 | $17\alpha(H),21\beta(H)$-(22R)-Homohopane |

*Data availability.* Raw data used in this study are archived at the Institute of Earth Environment,
Chinese Academy of Sciences, and are available on request by contacting the corresponding
author.
*Supplement.* The Supplement related to this article is available online at
*Author contributions.* RJH designed the study. Data analysis was done by WY, LY, and RJH.
WY, LY and RJH interpreted data, prepared the display items and wrote the manuscript. All
authors commented on and discussed the manuscript.
*Acknowledgements.* This work was supported by the National Natural Science Foundation of
China (NSFC) under grant no. 41877408 and no. 91644219, the Chinese Academy of Sciences
(no. ZDBS-LY-DQC001), the Cross Innovative Team fund from the State Key Laboratory of
Loess and Quaternary Geology (SKLLQG) (no. SKLLQGTD1801), and the National Key
Research and Development Program of China (no. 2017YFC0212701). Yongjie Li
acknowledges funding support from the National Natural Science Foundation of China



(41675120), the Science and Technology Development Fund, Macau SAR (File no. 016/2017/A1), and the Multi-Year Research grant (No. MYRG2018-00006-FST) from the University of Macau.

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





**Table 1.** Annual and seasonal mean contributions of measured PAHs, NACs and MOPs to
methanol-soluble BrC light absorption at 365 nm. Hyphens denote the measured value of more
than one third of the samples is below the detection limit.

| Compounds | MAE$_{365}$ (m$^2$ g$^{-1}$) | Contribution to BrC light absorption at 365 nm (%) | | | | |
|---|---|---|---|---|---|---|
| | | Annual | Spring | Summer | Fall | Winter |
| Fluoranthene (FLU) | 4.25 | 0.11 | 0.05 | 0.02 | 0.05 | 0.15 |
| Pyrene (PYR) | 0.46 | 0.01 | 0.00 | 0.00 | 0.01 | 0.01 |
| Chrysene (CHR) | 0.00 | 0.00 | 0.00 | 0.00 | 0.00 | 0.00 |
| Benzo(a)anthracene (BaA) | 2.06 | 0.04 | 0.01 | 0.01 | 0.02 | 0.05 |
| Benzo(a)pyrene (BaP) | 9.31 | 1.04 | 0.76 | 0.39 | 1.16 | 1.10 |
| Benzo(b)fluoranthene (BbF) | 4.10 | 0.17 | 0.14 | 0.07 | 0.17 | 0.18 |
| Benzo(k)fluoranthene (BkF) | 3.47 | 0.04 | 0.03 | 0.02 | 0.04 | 0.04 |
| Indeno[1,2,3-cd]pyrene (IcdP) | 4.68 | 0.51 | 0.50 | 0.24 | 0.71 | 0.46 |
| Benzo(ghi)perylene (BghiP) | 8.95 | 0.29 | 0.28 | 0.16 | 0.41 | 0.26 |
| 9,10-Anthracenequinone (9,10AQ) | 0.28 | 0.01 | 0.00 | 0.00 | 0.00 | 0.01 |
| Benzanthrone (BEN) | 6.13 | 0.11 | 0.08 | 0.05 | 0.11 | 0.12 |
| Benzo[b]fluoren-11-one (BbF11O) | 1.89 | 0.02 | 0.02 | 0.01 | 0.02 | 0.03 |
| 4-Nitrophenol (4NP) | 2.17 | 0.08 | 0.06 | 0.02 | 0.05 | 0.10 |
| 4-Nitro-1-naphthol (4N1N) | 9.71 | - | - | - | - | 0.03 |
| 2-Methyl-4-nitrophenol (2M4NP) | 2.81 | 0.03 | 0.01 | 0.01 | 0.01 | 0.04 |
| 3-Methyl-4-nitrophenol (3M4NP) | 2.65 | 0.02 | 0.01 | 0.00 | 0.01 | 0.03 |
| 2,6-Dimethyl-4-nitrophenol (2,6DM4NP) | 3.27 | - | - | - | - | 0.01 |
| 4-Nitrocatechol (4NC) | 7.91 | 0.27 | 0.05 | 0.03 | 0.20 | 0.35 |
| 3-Methyl-5-nitrocatechol (3M5NC) | 5.77 | - | - | - | 0.05 | 0.11 |
| 4-Methyl-5-nitrocatechol (4M5NC) | 7.29 | - | - | - | 0.06 | 0.13 |
| 3-Nitrosalicylicacid (3NSA) | 3.86 | - | - | - | - | 0.01 |
| 5-Nitrosalicylicacid (5NSA) | 3.36 | 0.03 | 0.01 | 0.02 | 0.04 | 0.02 |
| Syringyl acetone (SyA) | 0.25 | 0.01 | 0.01 | 0.00 | 0.01 | 0.01 |
| Vanillin (VAN) | 8.17 | 0.01 | 0.00 | 0.00 | 0.00 | 0.01 |
| Vanillic acid (VaA) | 0.66 | 0.00 | 0.00 | 0.00 | 0.00 | 0.00 |
| Total | 103.46 | 2.80 | 2.02 | 1.05 | 3.13 | 3.26 |



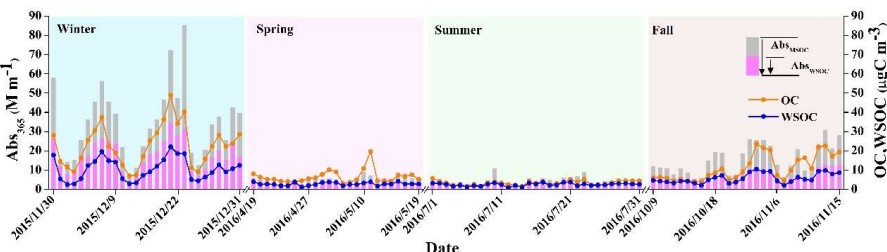


**Figure 1.** Time series of the light absorption coefficient of water-soluble and methanol-soluble BrC at 365 nm (Abs$_{365,WSOC}$ and Abs$_{365, MSOC}$, respectively), as well as OC and WSOC concentrations.






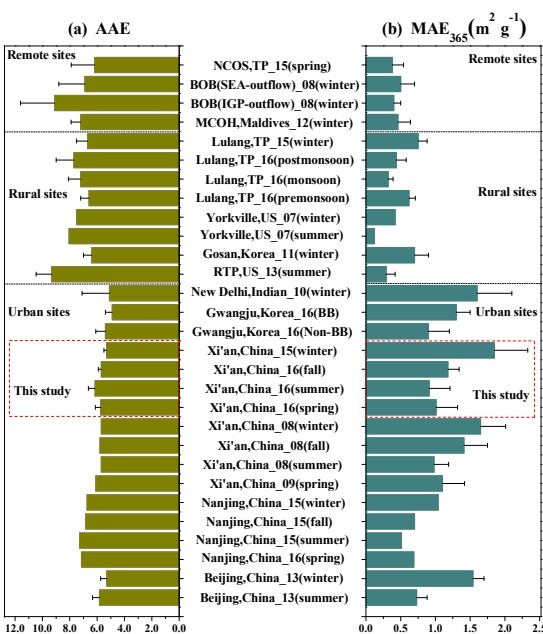


**Figure 2.** Comparison of AAE (left column) and MAE$_{365}$ (right column) values of water-soluble

BrC at remote sites (Srinivas and Sarin, 2013; Bosch et al., 2014; Zhang et al., 2017b), rural

sites (Hocobian et al., 2010; Kirillova et al., 2014a; Zhu et al., 2018; Xie et al., 2019) and urban

sites (Kirillova et al., 2014b; Yan et al., 2015; Chen et al., 2018; Huang et al., 2018; Park et al.,

2018).



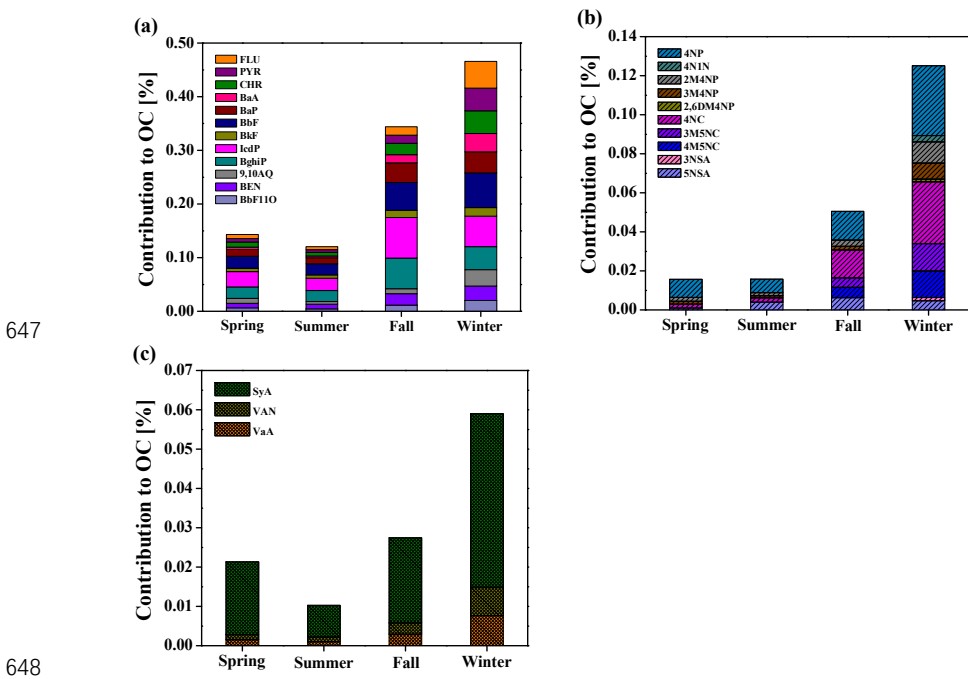

**Figure 3.** Contributions of (a) PAHs, (b) NACs, and (c) MOPs carbon mass concentrations to the total OC concentrations.









**(c)**


**(d)**


**Figure 4.** Light absorption contributions of (a) PAHs, (b) NACs, (c) MOPs and (d) total






measured chromophores to $Abs_{MSOC}$ over the wavelength range of 300 to 500 nm in spring,
summer, fall and winter.





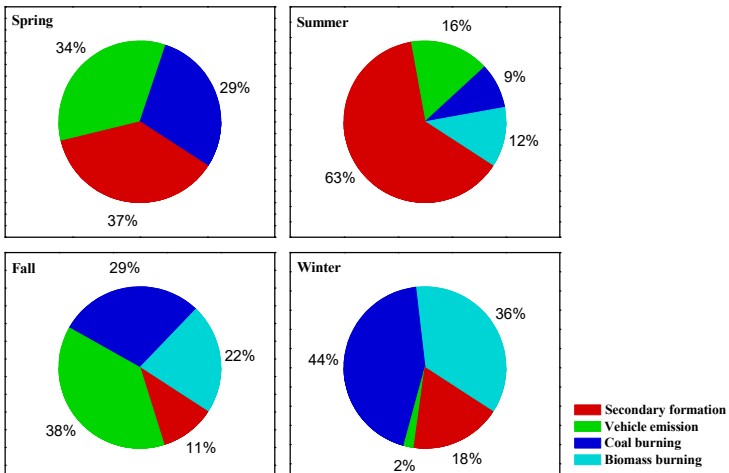


**Figure 5.** Contributions of the major sources to $Abs_{365,MSOC}$ in Xi'an during spring, summer, fall

and winter.