# Peer review of "Characterization of the light absorbing properties, chromophores composition"

_Atmospheric Chemistry and Physics, 2019_

## Referee Comment (RC1) · Anonymous Referee #1 · 3 Jan 2020

Brown carbon (BrC) is a fraction of organic aerosols with effective light absorption, which has significant effects on radiative forcing and climate. In the present study, the light absorbing properties, chromophores composition, and sources of BrC were investigated for aerosols collected in Xi'an, Northwest China. The results showed that AAE and MAE365 both present distinct seasonal differences and were due to the differences in sources and chemical composition of BrC chromophores. Some organic compounds including 12 PAHs, 10 NACs and 3 MOPs were quantified, which contributions to the light absorption of methanol-soluble BrC light absorption at 365 nm ranged from 1.1% to 3.3%, and thereby indicates that the light absorption of BrC is likely determined by an amount of chromophores with strong light absorption ability. Four major sources of methanol-soluble BrC were identified by PMF, which including secondary formation, vehicle emission, coal combustion and biomass burning and a large variation of BrC sources was observed in different seasons. Overall the manuscript is written well, and with some further explanation of collected data and further elaboration on the results it will be ready for publication. Below are specific revision comments for the authors to consider in their next revision:

1) Line 113: Please provide the unit of $Abs\lambda$.

2) Line 122: Please provide the unit of $MAE_{365}$.

3) Line 126: "MOSC" should be "MSOC".

4) Lines 139-140: "The concentrations of NACs were analyzed following the method by Al-Naiema and Stone (2017). Briefly……". The details of experiment have some differences to that of reference (Al-Naiema and Stone, 2017). For example, the silylation was conducted by heating at 70 ℃ for 3h in this study, however it was conducted by heating at 100 ℃ for 90 min in the reference (Al-Naiema and Stone, 2017). In addition, according to the reference (Al-Naiema and Stone, 2017), the derivatization method used in the current study is only used for levoglucosan and phthalic acid isomers. Please check this section.

5) However about the uncertainty of organic compounds and PMF analysis?

6) Lines 179-183: As shown in the paper "The higher WSOC fraction in OC during summer may

be related to biomass burning emissions…? Why biomass burning have a large emissions in summer? The seasonal variation of biomass burning should be small.

"The lower WSOC fractions in OC during winter could be attributed to enhanced emissions from coal combustion and motor vehicles": I think the seasonal variation of motor vehicles emissions should be very small.

This explanation of seasonal variations of WSOC/OC should be revised based the experimental results and the supporting references.

7) Lines 212-215: the average MAE365 value (1.18) in fall is more similar to that in spring and summer.

8) Lines 218-220: How about the contribution of the large amount of coal combustion and biomass burning activities in rural region around Xi`an?

9) Line 212-216: The unit of $MAE_{365}$ is $m^2\ gC^{-1}$, however the unit of $MAE_{365}$ is $m^2\ g^{-1}$ in Fig 2 and S2, Table 1. Please correct the errors. This is also important for the calculation of light absorption contribution of various organic compounds.

10) Lines 77-78: Other important references about BrC materials directly emitted from coal combustion should added, such as "Sun et al., ACP, 2017, 17, 4769", "Li et al., EST 2019, 53, 595", "Song et al., EST 2019, 53, 13607", etc.

11) Line 247: The "autumn" should be revised to "fall".

12) The PAHs, NACs and MOPs are important strong light-absorbing organic compounds, however the total contributions of PAHs, NACs and MOPs to the light absorption of methanol soluble BrC at 365 nm are small, only 1.05%- 3.26%. What is the major contribution to the light absorbing BrC?

13) Section 3.3: the sources of BrC were quantified with a PMF model. However I have several concerns: 1) Why the contribution of biomass burning was not identified in spring? In general, the biomass burning activities should happen in every seasons.2) the contribution of SOA is lowest in Fall. Why? Could you give some explaination? 3) the contribution of vehicle emissions are more than 1/3 in spring and fall. Could you give some discussion to interpret the reason for this seasonal variations of source compostions.

---

## Referee Comment (RC2) · Anonymous Referee #2 · 9 Jan 2020

In this work, the authors investigated the optical properties, chemical composition and sources of brown carbon (BrC) in Xi'an from 2015-2016. They identified three groups of BrC chromophores including PAHs and their derivatives, nitrophenols and methoxyphenols, of which some were not identified as BrC chromophores in previous studies (e.g., methoxyphenols). The authors then quantified the contribution of these identified chromophores to the total light absorption of BrC at the wavelength from 300-500 nm, which is important dataset because the link between BrC absorption and chemical composition is a key for estimating the effect of BrC on radiative forcing but such data are still very limited particularly for ambient measurements. Finally, the authors quantified the sources of BrC by PMF using these identified chromophores and found the seasonal

difference in the contributing sources. In general, the results are provided in a concise format and the discuss is well stated and directly related to the important aspects of BrC, i.e., the links between optical properties, chromophore composition, and sources of BrC. Also, the paper is well written and organized. I recommend publication in ACP after minor revision.

Specific comments

1. The peak values of the light absorption contribution of the measured chromophores are not always at 365 nm. Therefore, it could be better to include the average light absorption contribution of these chromophores to BrC at the wavelength of 300-500 nm.

2. Previous studies often discussed the light-absorption contribution of chromophores to water-soluble BrC. The authors discussed only the contribution to methanol-soluble BrC in this study. Should they also discuss the contribution to water-soluble BrC from the identified chromophores?

3. Page 6, line 162. Change "9,10-anthracenequinone (9,10-AQ)" to "9,10-anthracenequinone (9,10 AQ)".

4. Page 6, line 163-166. Not all species are non-light absorbing. For example, picene contains five benzene ring and should be light-absorbing species. It could be better to change "non-light absorbing markers" to "commonly used markers".

5. Page 11, line 300-301. 9,10 AQ, BEN and BbF11O are not only from combustion emission but also from secondary formation. Please clarify it.

6. Page 26. Figure 2. Change m2 g-1 to m2 gC-1.

---

## Referee Comment (RC3) · Anonymous Referee #3 · 12 Jan 2020

This manuscript describes how different organic compounds contribute to the absorption properties of ambient aerosols in Xi'an (Northwest China). PM2.5 samples were collected during all four seasons and analyzed for optical properties (spectrophotometer measurements), total organic carbon (TOC), 12 polycyclic aromatic hydrocarbons (PAHs), 10 nitrated aromatic compounds (NAC), 3 methoxyphenols, and 4 hopanes. Prior to the analyses, the filters were extracted with water and methanol. The aim of this study was to estimate the contribution of BrC species to the optical properties of ambient PM2.5. This study is scientifically important. The manuscript is well organized and well written. However, there are four major comments.

[Figure]

Major comments:

1. The author extracted and analyzed many non-polar organic compounds (PAHs, hopanes, etc.). However, for the extraction, solvents with high polarity indexes were used (water and methanol). By using these solvents, the author would not be able to extract non-polar compounds and estimate their contribution to the non-polar BrC fraction of the collected PM2.5. Sengupta et al. (2018) highlighted the importance of the non-polar fraction of BrC aerosols. Plus, the reference to this study is missing.

2. Many organic species from different glasses and with different volatility levels were measured. However, only one deuterated internal standard (4-nitrophenol-d4) was used to account for potential losses of analytes during the extraction and pre-concentration procedures. How were losses of other organic species (besides 4-nitrophenol) taken into account?

3. It was highlighted that different sources make different contributions to the chemical composition of PM2.5 collected in Xi'an. At the same time, the discussion (description) of these sources (how far they are from the sampling site, meteorological conditions, transport, types of biomass-burning fuels, etc.) is missing. Therefore, it is very hard to evaluate what composition of PM2.5 should be expected.

4. Lines 304–310. References and data are missing on four used factors of the source apportionment.

Some minor comments:

Line 55. References are needed on adverse health effects of PAHs.

Lines 99, 108, 112, 139. Company name (+city, state, country) of material and instruments is missing.

Line 149. What is the company (+city, country, etc.) of the GC column?

Line 204. References on absorption properties (above 300 nm) of PAHs are needed.

[Figure]

Line 215. It should be specified that "such large seasonal differences indicate seasonal difference in BrC sources" for the Xi'an area (Northwest China). Again, a good description of these sources is needed in the manuscript.

In summary, I recommend this manuscript for publication after the author addresses the major questions.

—————————————————————

---

## Author Comment (AC1) · 14 Mar 2020

The authors thank the referees to review our manuscript and particularly for the valuable comments and suggestions that have significantly improved the manuscript. We provide below point-by-point responses (in blue) to the referees' comments and have made changes accordingly in the revised manuscript.

Referee #1

Brown carbon (BrC) is a fraction of organic aerosols with effective light absorption, which has significant effects on radiative forcing and climate. In the present study, the light absorbing properties, chromophores composition, and sources of BrC were investigated for aerosols collected in Xi'an, Northwest China. The results showed that AAE and MAE365 both present distinct seasonal differences and were due to the differences in sources and chemical composition of BrC chromophores. Some organic compounds including 12 PAHs, 10 NACs and 3 MOPs were quantified, which contributions to the light absorption of methanol-soluble BrC light absorption at 365 nm ranged from 1.1% to 3.3%, and thereby indicates that the light absorption of BrC is likely determined by an amount of chromophores with strong light absorption ability. Four major sources of methanol-soluble BrC were identified by PMF, which including secondary formation, vehicle emission, coal combustion and biomass burning and a large variation of BrC sources was observed in different seasons. Overall the manuscript is written well, and with some further explanation of collected data and further elaboration on the results it will be ready for publication. Below are specific revision comments for the authors to consider in their next revision:

Specific comments

1) Line 113: Please provide the unit of Abs$\lambda$.

Response: The unit of Abs$_\lambda$ (M m$^{-1}$) have been provided.

2) Line 122: Please provide the unit of MAE365.

Response: The unit of MAE$_{365}$ (m$^2$ gC$^{-1}$) have been provided.

3) Line 126: "MOSC" should be "MSOC".

Response: Change made.

4) Lines 139-140: "The concentrations of NACs were analyzed following the method by Al-Naiema and Stone (2017). Briefly……". The details of experiment have some differences to that of reference (Al-Naiema and Stone, 2017). For example, the silylation was conducted by heating at 70 °C for 3h in this study, however it was conducted by heating at 100 °C for 90 min in the reference (Al-Naiema and Stone, 2017). In addition, according to the reference (Al-Naiema and Stone, 2017), the derivatization method used in the current study is only used for levoglucosan and phthalic acid isomers. Please check this section.

Response: Thanks for your careful reading. The silylation reaction with BSTFA at 70 °C for 3 h is a routine derivatization method for polar organic species before GC-MS analysis (e.g., Wang et al., 2006; Al-Naiema and Stone, 2017). In Al-Naiema and Stone (2017), the derivatization was conducted by heating to 70 °C for 3 h for levoglucosan and phthalic acid isomers, but modified slightly by heating to 100 °C for 90 min for nitromonoaromatics to get more symmetrical peak shapes and higher intensities than the derivatization method used for levoglucosan and phthalic acid isomers. In our study, however, with the routine method of "70 °C for 3 h" we also got symmetrical peak shapes and high intensities for NACs (see Figure below), and both NACs and other organic compounds can be simultaneously analyzed.

In line 144-162, we have changed "The concentrations of NACs were analyzed…following methods described by Wang et al. (2006)" to "Prior to the GC-MS analysis, the silylation derivatization was conducted using a routine method (e.g., Wang et al., 2006; Al-Naiema and Stone, 2017). Briefly, a quarter of…and a GC inlet of 280 °C. The GC oven temperature was held at 50 °C for 2 min, ramped to 120 °C at a rate of 15 °C

min$^{-1}$, and finally reached 300 °C at a rate of 5 °C min$^{-1}$ (held for 16 min). Note that the derivatization for NACs was conducted at 70 °C for 3 h which is slightly different from the protocol used in Al-Naiema and Stone (2017), because symmetrical peak shapes and high intensities for NACs can also be obtained under this condition in our study (see Fig. S1).

[Figure]

**Figure S1.** Selected ion monitoring chromatograms (GC-MS) for nitrated aromatic compound (NAC) standards (2 ug mL$^{-1}$) measured in our study.

5) How about the uncertainty of organic compounds and PMF analysis?

Response: We re-checked the uncertainties (RSDs) and have now added these values in the revised manuscript.

In line 167, it now reads "…the uncertainties (RSDs) are < 10% for measured organic compounds..

In line 335-336, it now reads "The uncertainties for PMF analysis are < 10% for secondary formation and biomass burning, < 15% for vehicle emission and coal burning."

6) Lines 179-183: As shown in the paper "The higher WSOC fraction in OC during summer may be related to biomass burning emissions…? Why biomass burning have a large emissions in summer? The seasonal variation of biomass burning should be

small.

"The lower WSOC fractions in OC during winter could be attributed to enhanced emissions from coal combustion and motor vehicles": I think the seasonal variation of motor vehicles emissions should be very small.

This explanation of seasonal variations of WSOC/OC should be revised based the experimental results and the supporting references.

Response: Daellenbach et al. (2016) reported that ~65% of biomass burning OA mass is water soluble, higher than cooking OA (~54%) and much higher than traffic OA (~11%). Therefore, we considered that biomass burning emissions, together with SOA, may contribute to higher WSOC fraction in summer, consistent with those reported in Ram et al. (2012) and Yan et al. (2015). The emissions of biomass burning indeed show large seasonal variation in northwest China (e.g., Xi'an). For example, Huang et al., (2018) reported the concentration of levoglucosan in Xi'an was about 11 times higher in winter than in summer because of large biomass burning for residential heating in winter. In summer, it is mainly from open burning of agricultural residues, e.g., wheats that were planted in previous winter and harvested in June/July.

To clarify this point, in 196-198, we changed "The higher WSOC...Yan et al., 2015)" to "The higher WSOC fraction in OC during summer is largely contributed by SOA and to some extent by biomass burning emissions because both SOA and biomass burning OA consist of high fraction of WSOC (Ram et al., 2012; Yan et al., 2015, Daellenbach et al., 2016)."

In winter, more water-insoluble organics are emitted by enhanced coal combustion for residential heating. We have changed "could be attributed to enhanced emissions from coal combustion and motor vehicles" to "could be attributed to enhanced emissions from coal combustion".

7) Lines 212-215: the average MAE365 value (1.18) in fall is more similar to that in spring and summer.

Response: Thanks for pointing this out. We have revised it to "...with highest values in winter (1.85 and 1.50 $m^2$ $gC^{-1}$, respectively), followed by fall (1.18 and 1.52 $m^2$ $gC^{-1}$), spring (1.01 and 0.79 $m^2$ $gC^{-1}$), and summer (0.91 and 1.21 $m^2$ $gC^{-1}$)."

8) Lines 218-220: How about the contribution of the large amount of coal combustion and biomass burning activities in rural region around Xi`an?

Response: The rural and remote sites in Fig. 2 refer to regions with less anthropogenic activities. We have clarified this point in the revised manuscript. In line 239-240, it now reads "...are obviously higher in urban sites than in rural and remotes sites that are less influenced by anthropogenic activities."

9) Line 212-216: The unit of MAE365 is m2 gC-1, however the unit of MAE365 is m2 g-1 in Fig 2 and S2, Table 1. Please correct the errors. This is also important for the calculation of light absorption contribution of various organic compounds.

Response: In Fig. 2 and S2, Table 1, we have changed the unit to $m^2$ $gC^{-1}$.

10) Lines 77-78: Other important references about BrC materials directly emitted from coal combustion should added, such as "Sun et al., ACP, 2017, 17, 4769", "Li et al., EST 2019, 53, 595", "Song et al., EST 2019, 53, 13607", etc.

Response: Thanks. We have now cited those references in the revised manuscript. In line 78-79, it now reads "...are also important primary sources of BrC (Sun et al., 2017; Yan et al., 2017; Xie et al., 2017; Li et al., 2019; Song et al., 2019)."

11) Line 247: The "autumn" should be revised to "fall".

Response: Change made.

12) The PAHs, NACs and MOPs are important strong light-absorbing organic compounds, however the total contributions of PAHs, NACs and MOPs to the light absorption of methanol soluble BrC at 365 nm are small, only 1.05%- 3.26%. What is the major contribution to the light absorbing BrC?

Response: This is indeed a very good question. As discussed in Laskin et al. (2015), our understanding of the BrC molecular composition and chemistry as well as the link with optical properties is still in its early stages. The light-absorbing contribution (at 365 nm) of the 25 chromophores measured in our study is small but comparable to those in previous studies (Mohr et al., 2013; Zhang et al., 2013; Teich et al., 2017; Huang et al., 2018). Also, the light absorption contribution is ~5 times higher than the carbon mass contribution to OC, indicating that these three groups of chromophores (PAHs, NACs and MOPs) are important components of BrC with high potential to absorb light on a same carbon mass basis.

Indeed, a large fraction of BrC chromophores are still not identified so far, and more studies are therefore necessary. Based on laboratory and ambient studies, more organics should be considered in future studies, including imidazoles (Kampf et al., 2012; Teich et al., 2016), quinones (Lee et al., 2014; Pillar et al., 2017), nitrogenous PAHs (Lin et al., 2016; Lin et al., 2018), polyphenols (Lin et al., 2016; Pillar et al., 2017) and oligomers with higher conjugation (Lin et al., 2014; Lavi et al., 2017).

We have added the following discussion. In line 308-314, it now reads "...with strong light absorption ability (Kampf et al., 2012; Teich et al., 2017). Of note, a large fraction of BrC chromophores are still not identified so far, and more studies are therefore necessary to better understand the BrC chemistry. Based on laboratory and ambient studies, imidazoles (Kampf et al., 2012; Teich et al., 2016), quinones (Lee et al., 2014; Pillar et al., 2017), nitrogenous PAHs (Lin et al., 2016; Lin et al., 2018), polyphenols (Lin et al., 2016; Pillar et al., 2017) and oligomers with higher conjugation (Lin et al., 2014; Lavi et al., 2017) could be included in future studies."

13) Section 3.3: the sources of BrC were quantified with a PMF model. However I have several concerns: 1) Why the contribution of biomass burning was not identified in spring? In general, the biomass burning activities should happen in every seasons. 2) the contribution of SOA is lowest in Fall. Why? Could you give some explaination? 3) the contribution of vehicle emissions are more than 1/3 in spring and fall. Could you give some discussion to interpret the reason for this seasonal variations of source compostions.

Response: We thank reviewer for raising these concerns, as we agree that further clarification will improve the manuscript. Here we provide responses to each of the question raised.

1) The biomass burning activities in Xi'an and surrounding areas were mainly in winter heating period and two harvest seasons (wheat in June and maize in Oct, respectively). Therefore, we believe that the biomass burning contribution in spring (April-May in our study) might be too small to be identified.

2) For the contribution of secondary formation to total $Abs_{365,MSOC}$, we ought to look at contributions from both relative and absolute terms. As shown in Table R1, the calculated absolute contributions of secondary formation to $Abs_{365,MSOC}$ were 1.75, 2.55, 1.70, 6.20 M m$^{-1}$ in spring, summer, fall, and winter, respectively. While the high contribution in winter can be attributed to abundant precursors (volatile organic compounds) co-emitting with the other primary sources (especially coal combustion and biomass burning), the high contribution in summer might be due to strong photochemical activity. For spring and fall, the absolute contributions from secondary formation were very similar, indicating moderate precursor emission and moderate photochemical activity. The low relative contribution of secondary formation to $Abs_{365,MSOC}$ in fall was due in part to the large contributions from primary emissions, e.g., coal burning (29% or 4.47 M m$^{-1}$) and biomass burning (22% or 3.39 M m$^{-1}$) that made up a total $Abs_{365,MSOC}$ in fall.

To avoid confusion, we replace Figure 5 with pie charts representing absolute contributions by the surface area of the sums of pies.

3) As shown in Table. R2, the seasonal differences of hopane concentrations (~10 times) was much smaller than those of PAHs (~30 times) and levoglucosan (~160 times), indicating that the differences of vehicle emission strength were relatively small among seasons. In summer, secondary formation contributed to over 60% of $Abs_{365,MSOC}$, although the total value of $Abs_{365,MSOC}$ was the smallest (4.05 M $m^{-1}$) among the four seasons. In winter, on the contrary, primary emissions from coal burning and biomass burning, other than vehicle emission, made up 80% of the total $Abs_{365,MSOC}$, which by itself was the highest (34.42 M $m^{-1}$) in the four seasons. Without as efficient secondary formation as in summer and as abundant other primary emissions as in winter, vehicle emission in spring and fall stood out as the significant contributor to $Abs_{365,MSOC}$. Note that the absolute contributions of vehicle emission to $Abs_{365,MSOC}$ were still higher in spring and fall than those in summer and winter (Table R1, or Table S4), yet these differences by a factor of 2–9 are still less pronounced than the differences (spring/fall vs winter) for other primary emissions (>40 times for coal burning and >25 times for biomass burning). Nevertheless, we agree with the reviewer that relative (and absolute) contribution of vehicle emission in fall was relatively higher, which might be affected by higher relative humidity in fall (on average 83% in fall vs. 61-69% in other seasons) resulting in higher vehicular $PM_{2.5}$ emissions (Chio et al., EPA, 2010). We have now added the following discussion in lines 349-363 the revised manuscript:

"…(wood and crop residues) and coal combustion for heating. In terms of absolute contributions to absorption of MSOC at 365 nm (see Table S4), secondary formation contributed 1.75, 2.55, 1.70, 6.20 M $m^{-1}$ in spring, summer, fall, and winter, respectively. The high contribution in winter can be attributed to abundant precursors (volatile organic compounds) co-emitted with other primary sources (especially coal burning and biomass burning), while the high contribution in summer might be due to strong photochemical activity. For spring and fall, the

absolute contributions from secondary formation were very similar, indicating moderate precursor emission and moderate photochemical activity. Also it should be noted that the absolute contributions of vehicle emission to absorption of MSOC at 365 nm were still higher in spring and fall than those in summer and winter, yet these differences by a factor of 2-9 are still less pronounced than the differences (spring/fall vs. winter) for other primary emissions (> 40 times for coal burning and > 25 times for biomass burning). In particular, the high vehicle contribution in fall might be affected by high relative humidity in fall (83% in fall vs. 61-69% in other seasons, on average) resulting in high vehicular $PM_{2.5}$ emissions (Chio et al., 2010). Such large seasonal difference…"

Table R1. Seasonal light absorption of methanol-soluble BrC at wavelength of 365 nm and the sources contributions.

|  | Spring | Summer | Fall | Winter |
|---|---|---|---|---|
| $Abs_{365,MSOC}$ $(M\ m^{-1})$ | 4.73 | 4.05 | 15.41 | 34.42 |
| | | | | |
| Sources contribution to $Abs_{365,MSOC}$ (%) | | | | |
| Secondary formation | 37 | 63 | 11 | 18 |
| Vehicle emission | 34 | 16 | 38 | 2 |
| Coal burning | 29 | 9 | 29 | 44 |
| Biomass burning | 0 | 12 | 22 | 36 |
| | | | | |
| Sources contribution to $Abs_{365,MSOC}$ $(M\ m^{-1})$ | | | | |
| Secondary formation | 1.75 | 2.55 | 1.70 | 6.20 |
| Vehicle emission | 1.61 | 0.65 | 5.86 | 0.69 |
| Coal burning | 1.37 | 0.36 | 4.47 | 15.41 |
| Biomass burning | 0 | 0.49 | 3.39 | 12.39 |

Table R2. Seasonal mean (± standard deviation) of the measured compounds.

| Compounds | Spring | Summer | Fall | Winter |
|---|---|---|---|---|
| *o*-ph | 3.92±2.29 | 6.50±3.57 | 8.70±5.04 | 11.19±7.56 |
| HP1 | 0.22±0.08 | 0.10±0.05 | 0.69±0.48 | 1.47±0.54 |
| HP2 | 0.23±0.11 | 0.11±0.06 | 0.66±0.44 | 1.15±0.41 |
| HP3 | 0.09±0.04 | 0.07±0.02 | 0.31±0.22 | 0.48±0.18 |
| HP4 | 0.09±0.03 | 0.07±0.02 | 0.27 ±0.19 | 0.55±0.27 |
| PI | 0.17±0.10 | - | 0.70±0.37 | 0.97±0.51 |
| FLU | 0.52±0.21 | 0.19±0.10 | 1.96±0.98 | 11.88±5.42 |
| PYR | 0.46±0.21 | 0.18±0.09 | 1.73±0.86 | 10.06±4.41 |
| CHR | 0.68±0.29 | 0.23±0.13 | 2.47±1.21 | 10.13±5.47 |
| BaA | 0.33±0.16 | 0.12±0.07 | 1.73±0.93 | 8.15±3.78 |
| BaP | 0.88±0.53 | 0.43±0.30 | 4.42±2.70 | 9.35±7.84 |
| BbF | 1.59±0.82 | 0.74±0.56 | 6.22±3.55 | 15.32±13.14 |
| BkF | 0.43±0.20 | 0.25±0.13 | 1.60±0.81 | 3.85±3.11 |
| IcdP | 2.02±1.19 | 0.84±0.43 | 9.22±4.89 | 13.46±12.37 |
| BghiP | 0.20±0.06 | 0.72±0.59 | 7.03±3.55 | 8.12±3.68 |
| 9,10AQ | 2.23±1.75 | 0.20±0.11 | 1.16±0.66 | 8.56±4.20 |
| BEN | 0.26±0.12 | 0.28±0.15 | 2.29±2.10 | 6.82±3.25 |
| BbF11O | 0.19±0.08 | 0.17±0.11 | 1.18±1.03 | 5.16±2.61 |
| LEV | 1.21±0.36 | 9.79±4.49 | 85.43±47.56 | 193.21±68.57 |
| VaA | 0.23±0.14 | 0.06±0.02 | 0.44±0.3 | 3.03±1.43 |
| VAN | 0.32±0.18 | 0.07±0.03 | 0.48±0.37 | 2.60±0.10 |
| SyA | 2.24±1.74 | 0.43±0.22 | 3.75±2.95 | 15.88±7.62 |

References

Li, M. J., Fan, X. J., Zhu, M. B., Zou, C. L., Song, J. Z., Wei, S. Y., Jia, W. L., and Peng, P. A.: Abundance and Light Absorption Properties of Brown Carbon Emitted from Residential Coal Combustion in China, Environ. Sci. Technol., 53, 595-603, 2019.

Song, J. Z., Li, M. J., Fan, X. J., Zou, C. L., Zhu, M. B., Jiang, B., Yu, Z. Q., Jia, W. L., Liao, Y. H., and Peng, P. A.: Molecular Characterization of Water- and Methanol-Soluble Organic Compounds Emitted from Residential Coal Combustion Using Ultrahigh-Resolution Electrospray Ionization Fourier Transform Ion Cyclotron Resonance Mass

Spectrometry, Environ. Sci. Technol., 53, 13607-13617, doi:10.1021/acs.est.9b04331, 2019.

Sun, J., Zhi, G., Hitzenberger, R., Chen, Y., Tian, C., Zhang, Y., Feng, Y., Cheng, M., Zhang, Y., Cai, J., Chen, F., Qiu, Y., Jiang, Z., Li, J., Zhang, G., and Mo, Y.: Emission factors and light absorption properties of brown carbon from household coal combustion in China, Atmos. Chem. Phys., 17, 4769–4780, doi:10.5194/acp-17-4769-2017, 2017.

Referee #2

In this work, the authors investigated the optical properties, chemical composition and sources of brown carbon (BrC) in Xi'an from 2015-2016. They identified three groups of BrC chromophores including PAHs and their derivatives, nitrophenols and methoxyphenols, of which some were not identified as BrC chromophores in previous studies (e.g., methoxyphenols). The authors then quantified the contribution of these identified chromophores to the total light absorption of BrC at the wavelength from 300-500 nm, which is important dataset because the link between BrC absorption and chemical composition is a key for estimating the effect of BrC on radiative forcing but such data are still very limited particularly for ambient measurements. Finally, the authors quantified the sources of BrC by PMF using these identified chromophores and found the seasonal difference in the contributing sources. In general, the results are provided in a concise format and the discuss is well stated and directly related to the important aspects of BrC, i.e., the links between optical properties, chromophore composition, and sources of BrC. Also, the paper is well written and organized. I recommend publication in ACP after minor revision.

Specific comments

1. The peak values of the light absorption contribution of the measured chromophores are not always at 365 nm. Therefore, it could be better to include the average light absorption contribution of these chromophores to BrC at the wavelength of 300-500 nm.

Response: Thanks for pointing it out. We have now added description of the average light absorption contribution of these chromophores to BrC at the wavelength of 300-500 nm.

In line 282-284, it now reads "The total contributions of PAHs, NACs and MOPs to the light absorption of methanol-soluble BrC ranged from 0.47% (summer) to 1.56% (winter) at the wavelength of 300-500 nm and ranged from 1.05% (summer) to 3.26% (winter) at the wavelength of 365 nm."

2. Previous studies often discussed the light-absorption contribution of chromophores to

water-soluble BrC. The authors discussed only the contribution to methanol-soluble BrC in this study. Should they also discuss the contribution to water-soluble BrC from the identified chromophores?

Response: Indeed, previous studies often discussed the light-absorption contribution of water-soluble chromophores (e.g., NACs) to water-soluble BrC (Zhang et al., 2013; Teich et al., 2017). However, in our study, we also quantified water-insoluble but methanol-soluble chromophores, e.g., PAHs. We believe that the methanol-soluble chromophores are under-represented, despite of a great deal of efforts spent on water-soluble chromophores. Therefore, we tend to focus on the contributions of these methanol-soluble chromophores to BrC.

3. Page 6, line 162. Change "9,10-anthracenequinone (9,10-AQ)" to "9,10-anthracenequinone (9,10 AQ)".

Response: Change made.

4. Page 6, line 163-166. Not all species are non-light absorbing. For example, picene contains five benzene ring and should be light-absorbing species. It could be better to change "non-light absorbing markers" to "commonly used markers".

Response: We have changed "non-light absorbing markers" to "commonly used markers" in the revised manuscript.

5. Page 11, line 300-301. 9,10 AQ, BEN and BbF11O are not only from combustion emission but also from secondary formation. Please clarify it.

Response: We have changed it. In line 328-331, it now reads "The inputs include vanillic acid, vanillin, and syringyl acetone for BrC from biomass burning, FLU, PYR, CHR, BaA, BaP, BbF, BkF, IcdP and BghiP, for BrC from incomplete combustion and other light absorbing chromophores 9,10AQ, BEN, and BbF11O."

6. Page 26. Figure 2. Change m2 g-1 to m2 gC-1.

Response: Chang made.

Referee #3

This manuscript describes how different organic compounds contribute to the absorption properties of ambient aerosols in Xi'an (Northwest China). PM2.5 samples were collected during all four seasons and analyzed for optical properties (spectrophotometer measurements), total organic carbon (TOC), 12 polycyclic aromatic hydrocarbons (PAHs), 10 nitrated aromatic compounds (NAC), 3 methoxyphenols, and 4 hopanes. Prior to the analyses, the filters were extracted with water and methanol. The aim of this study was to estimate the contribution of BrC species to the optical properties of ambient PM2.5. This study is scientifically important. The manuscript is well organized and well written. However, there are four major comments.

In summary, I recommend this manuscript for publication after the author addresses the major questions.

Major comments:

1. The author extracted and analyzed many non-polar organic compounds (PAHs, hopanes, etc.). However, for the extraction, solvents with high polarity indexes were used (water and methanol). By using these solvents, the author would not be able to extract non-polar compounds and estimate their contribution to the non-polar BrC fraction of the collected PM2.5. Sengupta et al. (2018) highlighted the importance of the non-polar fraction of BrC aerosols. Plus, the reference to this study is missing.

Response: The organic compounds quantified in our study were extracted by a mixture of dichloromethane/methanol (2:1, v/v) which can extract both polar and non-polar organic compounds (e.g., PAHs, hopanes, levoglucosan). However, the light absorption was measured by extracting BrC into methanol because methanol can extract ~90% of OC for ambient aerosol (e.g., Chen and Bond, 2010; Cheng et al., 2016; Xie et al., 2019) and has been widely used for BrC extraction (e.g., Cheng et al., 2016; Huang et al., 2018; Zhu et al., 2018). Meanwhile, these 25 organic compounds including the PAHs can all be dissolved in methanol, as we did for their standards.

We agree that non-polar fraction of BrC is important and the reference has been added.

In line 204-206, it now reads"…the optical properties of BrC could be largely underestimated when using water as the extracting solvent as non-polar fraction of BrC is also important to light absorption of BrC (Sengupta et al., 2018)."

2. Many organic species from different glasses and with different volatility levels were measured. However, only one deuterated internal standard (4-nitrophenol-d4) was used to account for potential losses of analytes during the extraction and preconcentration procedures. How were losses of other organic species (besides 4-nitrophenol) taken into account?

Response: In our study, 4-nitrophenol-2,3,5,6-d4 was used as an internal standard to correct for potential loss for NACs quantification (Chow et al., 2015). For the quantification of other organic compounds, an external standard method was used through daily calibration with working standard solutions. Also, for every 10 samples, a procedural blank and a spiked sample (i.e., ambient sample spiked with known amounts of standards) were measured to check the interferences and recoveries. The measured recoveries are 80-102% for measured organic compounds….We have added this description in the revised manuscript.

3. It was highlighted that different sources make different contributions to the chemical composition of PM2.5 collected in Xi'an. At the same time, the discussion (description) of these sources (how far they are from the sampling site, meteorological conditions, transport, types of biomass-burning fuels, etc.) is missing. Therefore, it is very hard to evaluate what composition of PM2.5 should be expected.

Response: The focus of this study was the seasonal differences in BrC optical properties (e.g., Abs, MAE), chromophore composition, and the sources. In particular, the BrC sources were resolved using these measured chromophores instead of commonly used non-light absorbing organic markers as model inputs, which can greatly

minimize the bias in quantifying the BrC sources using non-light absorbing markers. A comprehensive characterization of the $PM_{2.5}$ composition was not the objective of this study. Certainly, it will be interesting to understand how the BrC is affected by e.g., meteorological conditions, types of biomass fuels, and the formation and transformation of optical properties and chemical composition during transport. However, each of these aspects require intensive studies in the future.

4. Lines 304–310. References and data are missing on four used factors of the source apportionment.

Response: The profiles for the four factors, which were resolved in our ME-2 model, are shown in Figure S3. These profiles (data) are not from literature. To make it clear, in the section "Source apportionment of BrC" we have added the following "…(see Table S2). This source apportionment protocol is very similar to our previous study (Huang et al., 2014)."

Some minor comments:

Line 55. References are needed on adverse health effects of PAHs.

Response: We have added references. In line 55-56, it now reads "…on human health (Bandowe et al., 2014; Shen et al., 2018)".

Lines 99, 108, 112, 139. Company name (+city, state, country) of material and instruments is missing.

Response: Company name (+city, state, country) of material and instruments have been added.

Line 101. "…quartz-fiber filters (20.3 × 25.4 cm, Whatman, QM-A, Clifton, NJ, USA)…"

Line 111-112. "…methanol (HPLC grade, J. T. Baker, Phillipsburg, NJ, USA)…"

Line 115-116. "…liquid waveguide capillary cell (LWCC-3100, World Precision Instrument, Sarasota, FL, USA )…"

Line 142-143. "…gas chromatograph-mass spectrometer (GC-MS, Agilent Technologies, Santa Clara, CA, USA)…"

Line 149. What is the company (+city, country, etc.) of the GC column?

Response: The company of the GC column has been added.

Line 155. "…DB-5MS column (Agilent Technologies, Santa Clara, CA, USA)…"

Line 204. References on absorption properties (above 300 nm) of PAHs are needed.

Response: A reference have been added, i.e., Samburova et al., 2016.

Line 215. It should be specified that "such large seasonal differences indicate seasonal difference in BrC sources" for the Xi'an area (Northwest China). Again, a good description of these sources is needed in the manuscript.

Response: We have added the following discussion in this paragraph. It now reads "…indicate seasonal difference in BrC sources. For example, contributions from coal combustion and biomass burning were much larger in winter than in other seasons due to large residential heating activities (also see Section 3.3 for more details)."

References

Bandowe, B. A. M., Meusel, H., Huang, R-J., Ho, K., Cao, J., Hoffmann, T., and Wilcke, W.: PM2.5-bound oxygenated PAHs, nitro-PAHs and parent-PAHs from the atmosphere of a Chinese megacity: Seasonal variation, sources and cancer risk assessment, Sci. Total

Environ., 473–474, 77–87, 2014.

Huang, R. J., Zhang, Y. L., Bozzetti, C., Ho, K. F., Cao, J. J., Han, Y. M., Daellenbach, K. R., Slowik, J. G., Platt, S. M., Canonaco, F., Zotter, P., Wolf, R., Pieber, S. M., Bruns, E. A., Crippa, M., Ciarelli, G., Piazzalunga, A., Schwikowski, M., Abbaszade, G., Schnelle-Kreis, J., Zimmermann, R., An, Z. S., Szidat, S., Baltensperger, U., El Haddad, I., and Prévôt, A. S. H.: High secondary aerosol contribution to particulate pollution during haze events in China, Nature, 514, 218–222, 2014.

Samburova, V., Connolly, J., Gyawali, M., Yatavelli, R. L. N., Watts, A. C., Chakrabarty, R. K., Zielinska, B., Moosmüller, H., and Khlystov, A.: Polycyclic aromatic hydrocarbons in biomass-burning emissions and their contribution to light absorption and aerosol toxicity, Sci. Total Environ., 568, 391-401, doi:10.1016/j.scitotenv.2016.06.026, 2016.

Sengupta, D., Samburova, V., Bhattarai, C., Kirillova, E., Mazzoleni, L., Iaukea-Lum, M., Watts, A., Moosmüller, H., and Khlystov, A.: Light absorption by polar and non-polar aerosol compounds from laboratory biomass combustion, Atmos. Chem. Phys., 18, 10849–10867, doi:10.5194/acp-18-10849-2018, 2018.

Shen, M. L., Xing, J., Ji, Q. P., Li, Z. H., Wang, Y. H., Zhao, H. W., Wang, Q. R., Wang, T., Yu, L. W., Zhang, X. C., Sun, Y. X., Zhang, Z. H., Niu, Y., Wang, H. Q., Chen, W., Dai, Y. F., Su, W. G., and Duan, H. W.: Declining Pulmonary Function in Populations with Long-term Exposure to Polycyclic Aromatic Hydrocarbons-Enriched $PM_{2.5}$, Environ. Sci. Technol., 52, 6610–6616, 2018.

---

## Author Response (AR2)

Dear Prof. Nizkorodov,

Thank you very much for your time, and particularly for your careful reading and valuable suggestion. We have made all changes accordingly. Thank you again!

Lines 33: those show -> compounds that have

Response: Change made.

Lines 35: those show -> compounds that have a

Response: Change made.

Sentence on line 42 in spring, in fall, in winter, in summer should be at the end of each statement

Response: Change made.

Line 71: some of the papers by Lin reported contribution of resolved compounds to the overall absorption coefficient that was considerably higher than reported here. It might be worth citing these numbers. For example, Lin (2017) attributes about 50% of the total absorption to nitroaromatics. Do you have any suggestions why his numbers could be so different from numbers reported here?

Response: This is indeed a very good question. Lin et al. (2016) reported that in biofuels burning samples (sawgrass, peat, ponderosa pine, and black spruce), 40-60% of the bulk BrC absorption in the wavelength range of 300-500 nm may be attributed to 20 strong chromophores. Also, Lin et al. (2017) reported that during the biomass burning event (Lag Ba'Omer, a nationwide bonfire festival in Israel), nitroaromatic compounds accounted for ~50% of the total absorption of water-soluble BrC. These numbers are considerably higher than that reported in our study. The differences could be due to that the samples measured in Lin et al. (2016; 2017) were from biofuels burning or biomass burning event with abundant emissions of aromatic compounds which can be transformed to nitroaromatics through chemical reactions (Harrison et al., 2005; Mazzoleni et al., 2007; Stockwell et al., 2014 ), whereas our samples were from urban air with mixed sources. In addition, Lin et al. measured the contribution not from the specific compounds but over 20 elemental formulas which could have dozens or even hundred of structures and we measured the contribution from 25 specific compounds. Furthermore, in Lin et al. (2017), they calculated the contribution to light absorption of water-soluble BrC and We calculated the contribution to methanol-soluble BrC which is ~2 times higher than light absorption of water-soluble BrC.

We have now cited these numbers from Lin et al. (2016, 2017). In line 72-77, it now reads "… Lin et al. (2016) reported that in biofuels burning samples (sawgrass, peat, ponderosa pine, and black spruce), about 40-60% of the bulk BrC absorption in the wavelength range of 300-500 nm may be attributed to 20 strong chromophores and in another study (Lin et al., 2017) they reported that nitroaromatic compounds accounted for ~50% of the total absorption of water-soluble BrC during the biomass burning event in a nationwide bonfire festival in Israel."

Line 164-165: are -> were

Response: Change made.

Line 178: LOD (limit of detection) is a more common term

Response: Change made.

Line 216: compounds with high conjugation degree and strong light-absorbing capability (e.g., PAHs) at longer wavelength-> conjugated compounds that absorb strongly at longer wavelengths (e.g., PAHs)

Response: Change made.

Figure 1 appears very grainy on my computer, I would recommend uploading a higher resolution version in the final revision

Response: Thank you for your suggestion. We have provided a new version with higher resolution.

Figure 2: the red box showing results of this study and black dotted lines separating different types of sites may be hard to see; it may be worth making them thicker

Response: Thank you for your suggestion. We have made the red box and black dotted lines thicker in Figure 2.

[revised manuscript text omitted]